# Nerve Injury-Induced γH2AX Reduction in Primary Sensory Neurons Is Involved in Neuropathic Pain Processing

**DOI:** 10.3390/ijms241210148

**Published:** 2023-06-15

**Authors:** Yan Zhang, Hao Gong, Ji-Shuai Wang, Meng-Na Li, De-Li Cao, Jun Gu, Lin-Xia Zhao, Xin-Dan Zhang, Yu-Tao Deng, Fu-Lu Dong, Yong-Jing Gao, Wen-Xing Sun, Bao-Chun Jiang

**Affiliations:** 1Institute of Pain Medicine and Special Environmental Medicine, Nantong University, Nantong 226019, China; 2Department of Molecular and Cellular Pharmacology, Miller School of Medicine, University of Miami, Coral Gables, FL 33136, USA; 3The 1st Clinical Department, China Medical University, Shenyang 110122, China; 4Department of Pathology, Medical School, Nantong University, Nantong 226001, China; 5Department of Nutrition and Food Hygiene, School of Public Health, Nantong University, Nantong 226019, China

**Keywords:** γH2AX, SNI, ATM, PP2A, neuropathic pain

## Abstract

Phosphorylation of the serine 139 of the histone variant H2AX (γH2AX) is a DNA damage marker that regulates DNA damage response and various diseases. However, whether γH2AX is involved in neuropathic pain is still unclear. We found the expression of γH2AX and H2AX decreased in mice dorsal root ganglion (DRG) after spared nerve injury (SNI). Ataxia telangiectasia mutated (ATM), which promotes γH2AX, was also down-regulated in DRG after peripheral nerve injury. ATM inhibitor KU55933 decreased the level of γH2AX in ND7/23 cells. The intrathecal injection of KU55933 down-regulated DRG γH2AX expression and significantly induced mechanical allodynia and thermal hyperalgesia in a dose-dependent manner. The inhibition of ATM by siRNA could also decrease the pain threshold. The inhibition of dephosphorylation of γH2AX by protein phosphatase 2A (PP2A) siRNA partially suppressed the down-regulation of γH2AX after SNI and relieved pain behavior. Further exploration of the mechanism revealed that inhibiting ATM by KU55933 up-regulated extracellular-signal regulated kinase (ERK) phosphorylation and down-regulated potassium ion channel genes, such as potassium voltage-gated channel subfamily Q member 2 (*Kcnq2*) and potassium voltage-gated channel subfamily D member 2 (*Kcnd2*) in vivo, and KU559333 enhanced sensory neuron excitability in vitro. These preliminary findings imply that the down-regulation of γH2AX may contribute to neuropathic pain.

## 1. Introduction

Neuropathic pain is an unbearable chronic pain with a very high incidence, resulting in the disability of the patients, and is one of the most significant health problems facing humanity today [1,2]. The pathogenesis of neuropathic pain is extraordinarily complex and is associated with increased excitability of the primary sensory neurons [3]. However, the specific biological mechanisms of increased neuronal excitability have not been elucidated. The membrane potential resetting in the dorsal root ganglion (DRG) neurons may be the root cause of the change in the excitability of the DRG neurons [4]. Previous research has found that epigenetic regulations, including histone acetylation and non-coding RNA regulation, affect neuropathic pain by altering the gene expression profiles of DRG and may bring new promise to treating neuropathic pain [5].

Histone modification is a frequently observed epigenetic alteration under pathological conditions [6]. G9a histone methyltransferase modulates neuronal excitability and pain behavior via down-regulating the expression of the potassium channel genes, including potassium voltage-gated channel subfamily A member 4, potassium voltage-gated channel subfamily D member 2 *(Kcnd2*), and potassium voltage-gated channel subfamily Q member 2 (*Kcnq2*) in DRG [7]. Histone H3 phosphorylation changes neuronal plasticity by affecting gene expression [8]. However, there are very few studies on histone phosphorylation involved in pain. H2AX is a variant of histone H2A and is prone to phosphorylation, especially in DNA double-strand breaks (DSBs) conditions [9]. Indeed, without DSBs, the phosphorylation of H2AX at serine 139 (γH2AX) will also change [10]. Considering that γH2AX is susceptible to dynamic modification after being affected by DNA damage signals, and optic nerve crush injury has been demonstrated to lead to DNA damage and up-regulates γH2AX [11], we speculated that γH2AX might be involved in nerve injury-induced neuropathic pain.

The level of γH2AX is regulated by ataxia telangiectasia mutated (ATM), ataxia telangiectasia and Rad3-related (ATR), or DNA-dependent protein kinase (DNA-PK), which catalyzes H2AX phosphorylation, and protein phosphatase 2A (PP2A), which catalyzes the dephosphorylation of γH2AX [12]. ATM and ATR are distributed in the nervous system and play essential roles in controlling synaptic transmission and excitatory–inhibitory balance [13], and PP2A plays crucial regulatory roles in physiological and pathological processes via governing the phosphorylation state of cellular proteins, including in neurodegenerative diseases [14]. Moreover, recent research found that ATM can also be activated by non-DNA damage stressors, such as oxidative stress [15]. However, little is known about the connections between ATM/ATR/PP2A, primary sensory neurons, and neuropathic pain.

In this study, we measured the expression and location of γH2AX and ATM in the DRG of mice that underwent spared nerve injury (SNI) and explored their potential roles in neuropathic pain development. This study demonstrated that γH2AX mainly presented in the DRG neurons nucleus in naive mice. Unexpectedly, γH2AX and ATM were persistently down-regulated after SNI. The inhibition of ATM by pharmacological approaches or siRNA decreased the pain sensation threshold to mechanical or thermal stimulus, whereas the knockdown of PP2A by the intrathecal injection of siRNA alleviated SNI-induced pain behavior and γH2AX down-regulation. Altogether, we provide new evidence for the involvement of γH2AX and associated regulating mechanisms in neuropathic pain processing.

## 2. Results

### 2.1. The Expression of γH2AX in DRG Was Decreased after SNI

To reveal the expression trend of γH2AX in DRG after SNI, the levels of γH2AX were detected 1, 3, 7, 14, and 21 days after SNI using immunofluorescence. The results show that γH2AX was decreased from the first day after SNI and remained so for 21 days compared with the sham group (Figure 1A,B). Western blotting confirmed the decreased γH2AX protein level in the DRG 3 days after SNI (Figure 1C,D). The location of γH2AX in DRG was analyzed via double-labeled immunofluorescence. In the sham group, the γH2AX was primarily localized in neurons, marked with β-tubulin III (TUJ1), but there were no obvious signals in macrophages or satellite glial cells labeled with ionized calcium-binding adapter molecule 1 (IBA1) or glutamine synthetase (GS) in the DRG (Figure 1E). However, the γH2AX was localized in neurons and macrophages in the DRG at 3 days after SNI (Figure 1E). To determine the specific distribution of γH2AX in DRG neurons, the DRG were double-labeled for γH2AX with large myelinated, small peptidergic, and small nonpeptidergic neuron markers NF200, IB4, and CGRP, respectively, using immunofluorescence. The findings indicate that γH2AX was co-labeled with neurofilament-200 (NF200), calcitonin gene-related peptide (CGRP), and isolectin B4 (IB4) in neurons (Figure 2A–C). The quantification analysis revealed that, in the sham group, 87.5% (483/552) of large myelinated neurons, 87.78% (194/221) of small peptidergic neurons, and 87.04% (262/301) of small nonpeptidergic neurons were found to be positive for γH2AX. This suggests that γH2AX is present in these three types of neurons in the DRG.

### 2.2. The Expression of γH2AX in the Spinal Cord Was Unchanged after SNI

The spinal cord (SC) plays a crucial role in pain modulation, acting as a critical relay between peripheral nociceptors and the higher centers of the brain [16]. Therefore, we characterized the expression and cell distribution of γH2AX in SC. After SNI, there was no notable difference in γH2AX levels between the ipsilateral spinal dorsal horn three days post-surgery and the contralateral side (Figure 3A,B). To uncover the distribution of γH2AX in SC, we utilized markers for neurons (Neuronal Nuclei, NeuN), astrocytes (glial fibrillary acidic protein, GFAP), and microglia (IBA1) to conduct a double immunofluorescence with γH2AX on SC from mice 3 days after SNI. As shown in Figure 3A, the coexistence of γH2AX was observed with NeuN and IBA1, but not obviously with GFAP in the spinal cord, suggesting that neurons and microglia are the main sources of γH2AX in the area.

### 2.3. The Expression of H2AX in DRG Was Down-Regulated after SNI

Considering the influence of H2AX on the level of γH2AX, we assessed the expression levels of H2AX in DRG tissue obtained from sham and SNI mice 3 days post-surgery. The immunofluorescence indicated that H2AX was co-localized with DAPI (Figure 4A), indicating that H2AX was mainly located in nucleus. Additionally, the fluorescence intensity of H2AX in DRG 3 days after SNI was significantly decreased compared with the sham group (Figure 4B). Consistent with the immunofluorescence result, the Western blot results confirmed the decrease in H2AX protein level (Figure 4C,D). Based on the results, it appears that the reduction in γH2AX in DRG due to SNI could be associated with a decrease in H2AX levels. To further investigate the localization of H2AX in DRG tissue, we performed the double-staining of H2AX with neuronal marker TUJ1, satellite cell marker fatty acid binding protein 7 (FABP7), and microglia marker IBA1. The results showed that H2AX was predominantly co-localized with TUJ1, rather than IBA1 or GFAP (Figure 4E), indicating that H2AX is mainly located in DRG neurons.

### 2.4. The Expression of ATM in DRG Was Decreased after SNI

The level of γH2AX is also regulated by its protein kinases and protein phosphatases, and ATM and ATR are the protein kinases responsible for increasing γH2AX levels [12]. To understand why γH2AX is reduced in DRG post-SNI, we initially measured the expression levels of ATM and ATR in the nervous system using qPCR. Figure 5A shows that the nervous system tissue exhibited higher levels of ATM expression compared to ATR. Therefore, we focused our investigation on studying ATM in the subsequent experiments. To begin with, we assessed the mRNA expression of ATM in DRG tissue using qPCR after SNI surgery. The results revealed a significant decrease in ATM expression 1, 3, and 10 days post-surgery compared to the naive group (Figure 5B). Moreover, to identify the location of ATM, we performed the double-staining of ATM with TUJ1, or with FABP7, or with IBA1 in DRG tissue samples obtained after the sham operation. The results show that ATM was primarily localized in neurons, but not in macrophages or satellite glial cells (Figure 5C,D). Additionally, to identify whether ATM was located within the nucleus of the neurons, we traced the outer boundary of the nucleus with a dotted line according to the immunofluorescence result of DAPI, which was used to label the nucleus. The results indicate that ATM was present in both the cytoplasm and nucleus of the DRG neurons (as shown in Figure 5C), suggesting that ATM may have a role in regulating the levels of γH2AX, which also resides in the nucleus of DRG neurons.

### 2.5. γH2AX and Neuropathic Pain Were Regulated by ATM

To reveal the role of ATM in the expression of γH2AX in neurons and the DRG, the levels of γH2AX were detected after inhibiting ATM using KU55933 [17]. Immunofluorescence staining was performed, and the results showed that γH2AX was significantly reduced after pre-incubation with 10 or 100 µM of KU55933 for 6 h in ND7/23 cells, as compared with the control group (Figure 6A,B). Similarly, the analysis of protein levels revealed that the presence of KU55933 led to a dramatic decrease in γH2AX levels (Figure 6C,D). Furthermore, in vivo experiments using the intrathecal injection of KU55933 demonstrated a dose-dependent induction of mechanical allodynia and thermal hyperalgesia on naive mice (Figure 6E). Moreover, the levels of γH2AX in DRG were substantially inhibited by the intrathecal injection of 100µM of KU55933 (Figure 6F,G). The intrathecal injection of ATM siRNA also reduced the pain threshold and ATM expression (Figure 6H,I). These results implied that ATM is responsible for γH2AX in DRG, and the inhibition of the ATM/γH2AX pathway enhanced pain sensitivity.

### 2.6. The Excitability of DRG Neurons Was Enhanced by Inhibition of ATM

Neuropathic pain is closely related to the increased excitability and spontaneous activity of DRG neurons [18]. Thus, the effect of ATM on the electrophysiological activity of DRG neurons was explored via patch clamp recordings. DRG neurons were cultured in vitro and incubated with ATM inhibitor KU55933 (25 μM) for 10–12 h. We compared the action potentials (Aps) of DRG neurons between the control group and the group incubated with KU55933. In responding to 100 pA, 200 pA, and 300 pA ramp currents, the numbers of APs in naive group neurons were 4.60 ± 0.40, 9.90 ± 0.72, and 12.20 ± 0.57, respectively, while the numbers of Aps in KU55933-incubated DRG neurons were 6.70 ± 0.30, 12.40 ± 0.52, and 14.60 ± 0.62, respectively (Figure 7A,B), suggesting that the excitability of DRG neurons was increased by the inhibition of ATM. We further compared the membrane capacitance (CM) and found no significant difference after incubation with KU55933 compared to the control group (Figure 7C).

### 2.7. Inhibition of ATM Activated ERK Phosphorylation and Affected Pain-Related Genes Expression

ERK/MAPK (mitogen-activated protein kinase) and AKT (protein kinase B) pathways are essential in mediating chronic pain [19,20]. The phosphorylation of ERK and AKT were detected by Western blotting after the intrathecal injection of ATM inhibitor for 6 h to explore their roles in the ATM inhibition-induced sensitization of neuropathic pain. The results show that ATM inhibited by intrathecal injection KU55933 (100 μM) notably up-regulated ERK phosphorylation but had no significant effect on the phosphorylation of AKT (Figure 8A,B), suggesting that ATM inhibited the ERK/MAPK pathway. Neuronal excitability is regulated by many genes, such as *Kcnq2*, *Kcnd2*, and opioid receptor mu 1 (*Oprm1*) [21,22,23], which blunts neurons excitability. The expression of *Kcnq2*, *Kcnd2*, and *Oprm1* was dramatically down-regulated by ATM inhibitor or siRNA (Figure 8C), implying that ATM-induced sensitization of neuralgia pain was associated with the down-regulation of pain-inhibiting genes’ expressions.

### 2.8. The Knockdown of PP2A Up-Regulated γH2AX and Inhibited Neuropathic Pain

PP2A is the protein phosphatase of γH2AX [24]. By detecting the expression profile of PP2A in the nervous system and immune system, we found that PP2A was expressed in DRG (Figure 9A). To determine the role of PP2A in SNI-induced neuropathic pain, the pain behavior was detected after the knockdown of PP2A using siRNA. The results show that SNI-induced mechanical allodynia and thermal hyperalgesia were significantly alleviated after the injection of PP2A siRNA from 6 to 24 h (Figure 9B). The PP2A siRNA significantly inhibited the expression of *Pp2a* at 24 h after intrathecal injection (Figure 9C). Additionally, the levels of γH2AX in DRG were up-regulated considerably after the knockdown of PP2A (Figure 9D,E). These results indicated the knockdown of PP2A alleviated SNI-induced neuropathic pain, and this may be associated with the up-regulation of γH2AX.

## 3. Discussion

γH2AX is a research hotspot in the study of genotoxic stress and associated human disease [25,26]. It is used as a biomarker of DNA DSBs and senescence. However, whether γH2AX is involved in pain modulation was still unclear. This study first reported that the levels of γH2AX in DRG were down-regulated, and that the H2AX and ATM in DRG were decreased after SNI, which may be responsible for the γH2AX down-regulation. Moreover, the knockdown of PP2A in DRG, which up-regulated γH2AX, alleviated the mechanical allodynia and heat hyperalgesia induced by SNI. Finally, and equally importantly, the present work reported that inhibiting ATM down-regulated γH2AX, affected pain-related gene expression, and enhanced neuronal excitability and pain sensation.

γH2AX is prone to occur in injured cells or cells in inflammatory conditions which produce abnormal toxic metabolites, such as ROS, that can directly cause DSB [27]. Peripheral nerve damage elevates oxidative stress, which can induce the expression of inflammatory factors, in DRG or in spinal cord microglia [28,29]. Therefore, peripheral nerve injury might lead to the up-regulation of γH2AX. Contrary to expectation, the expressions of γH2AX and H2AX were down-regulated in DRG after SNI, a finding which was not consistent with the previous report that γH2AX in L4–L5 DRG was significantly increased after chronic constriction injury in rats [30]. These contradictory results may be due to the different models used and different times used for detection in these neuropathic pain studies.

ATM is a 370 kD serine/threonine kinase that is activated by DNA damage [31], and ATM triggers DNA repair by phosphorylating critical protein substrates after DNA damage [17,32]. It has been reported that ATM appears to be the major kinase associated with γH2AX [33]. In this study, ATM expression was consistently down-regulated after SNI, which is consistent with the trend of γH2AX changes. The in vitro and in vivo application of ATM inhibitor KU-55933 down-regulated the levels of γH2AX in ND7/23 cells and the DRG. Therefore, we consider ATM to be one of the reasons for the down-regulation of γH2AX expression in DRG after SNI; however, the upstream signaling pathway regulating ATM after peripheral nerve injury remains unclear and further investigation into this matter is needed. PP2A is one of the most abundant enzymes in the organism, accounting for 1% of the total cellular protein in some tissue [34], and mediates the dephosphorylation of γH2AX [12]. We found that the knockdown of PP2A with siRNA enhanced γH2AX levels, suggesting that it may be involved in the regulation of γH2AX decline in neurons under conditions of nerve injury.

γH2AX is associated with many neurologic disorders, including Alzheimer’s disease, Parkinson’s disease, and Huntington’s disease [26]. However, the role and mechanism of γH2AX in the occurrence and maintenance of pain have not been reported in detail. Our study found that the inhibition of γH2AX by inhibiting ATM or the knockdown of ATM led to the down-regulation of mechanical and thermal nociceptive behavior in naive mice. In contrast, the inhibition of PP2A by siRNA to increase the γH2AX level attenuates SNI-induced pain behaviors. These findings suggest that ATM/PP2A/γH2AX participates in the regulation of neuropathic pain. γH2AX is a post-translational modification of the histone. Thus, its down-regulation perhaps regulates the expressions of pain-related genes. Many potassium channel-coding genes [35] and *Oprm1* [36] are down-regulated in DRG after peripheral nerve injury and are involved in the increased excitability of primary sensory neurons and pain genesis and maintenance. Our data show that the deregulation of γH2AX by the ATM inhibitor or siRNA reduced the expression of *Kcnq2*, *Kcnd2*, and *Oprm1* in naive mice DRG and enhanced the excitability of the cultured primary sensory neuron. This observation implies that ATM may exert an analgesic effect by maintaining pain-repressing gene expression, and the relationship between γH2AX and Kcnq2, Kcnd2, and Oprm1 needs further study. 

Ionizing radiation causes γH2AX accumulation in neurons [37] and the level of γH2AX was increased in astrocytes in Alzheimer’s disease [38]. However, our study found that γH2AX is located in neurons and macrophages in DRG, and in neurons and microglial in SC, implying that SNI-induced DNA damage mainly occurred in neurons and macrophages or microglia, and not in astrocytes. DNA damage promotes the expression of genes related to inflammation [39,40]. To find the answer as to whether DNA damage affects inflammation-related gene expression in DRG macrophages or in SC microglia, which release of inflammation cytokines in neuropathic pain [41], further study is needed.

## 4. Materials and Methods

### 4.1. Animals and Surgery

ICR and C57BL/6 mice (4–6 weeks, male) were purchased from the Experimental Animal Center of Nantong University. All mice used were raised in a 12:12 light–dark period and had free availability of food and water. During the SNI surgery, we carefully ligated with 6–0 silk and transected the common peroneal and tibial nerves, and the sural nerve was preserved intact [42]. The mice in the sham-operated group were operated upon as described above, except that they were not ligated and transected. The Ethics Committee of Nantong University (S20230420-003) also reviewed and approved the animal study.

### 4.2. Drugs and Administration

KU-55933 (ATM Kinase Inhibitor, S1092) was purchased from Selleck. For intrathecal injection, siRNA was purchased from Suzhou GenePharma. PP2A-mus-705 sense 5′-3′: GGC AGA UCU UCU GUC UAC ATT, antisense 5′-3′: UGU AGA CAG AAG AUC UGC CTT. PP2A siRNA was mixed with TurboFect in vivo Transfection Reagent (Thermo Scientific, Waltham, MA, USA): siRNA/Transfection Reagent = 1:0.12. 

### 4.3. Behavioral Testing

From 3 days before the experiment, the mice were allowed to adapt to the experimental environment. The temperature and humidity of the room were stable for all of the experiments. During all of the behavioral experiments, the experimenter was blinded (unaware of the grouping of mice).

#### 4.3.1. Von Frey Filament Test

Before testing, the mice were placed in an acrylic mesh box and given about 45 min of habituation training. The metatarsal surface of the left hind paw was stimulated with a range of von Frey filaments of increasing logarithmic stiffness (0.02 g–2.56 g, Stoelting, Wood Dale, IL, USA). If the mice reacted negatively, the next, larger von Frey filament was applied. If the mice reacted positively, the next, smaller von Frey filament was applied. Dixon’s up-and-down method was used to identify the 50% paw withdrawal threshold.

#### 4.3.2. Hargreaves Test

Before the experiment, the mice were placed in an acrylic mesh box on a glass plate and given about 30 min to acclimate. Subsequently, the plantar surface of the foot was exposed to a beam of radiant heat through a transparent glass surface, and the latency of the paw withdrawal was recorded. The baseline latency was adjusted to 10–14 s, and the irradiation cutoff was set to 20 s to prevent potential thermal injury. Three trials were undertaken with an interval of 5 min each [43].

### 4.4. ND7/23 Cells Culture

ND7/23 cells were cultured in high-glucose Dulbecco’s modified Eagle’s medium (DMEM, Corning, NY, USA) containing 10% fetal bovine serum and 1% penicillin-streptomycin. The experiments were performed when the cells had grown to 70–90% confluence. For the inhibition of the ATM experiment, prior to stimulation with 10 and 100 μM of KU55933, high glucose DMEM was replaced with Opti-MEM without fetal bovine serum (Corning, NY, USA), and ND7/23 cells were incubated with or without KU55933 for 6 h. After treatment, the cells were collected for Western blotting or immunocytochemistry.

### 4.5. Immunohistochemistry

The mice were anesthetized with isoflurane, and 0.01 M PBS and 4% paraformaldehyde were injected into the ascending aorta. After the perfusion, the L3–L5 DRG and SC segment were removed, postfixed overnight in 4% PFA, gradually dehydrated with 20% and 30% sucrose, and then embedded in OCT. As described above, DRG (14 µm) sections or SC (30 µm, free-floating) were cut on the quick-freezing table and stained with immunofluorescence. In short, the tissue sections were first permeated and sealed for 2 h with 1% BSA and then incubated overnight with the antibody at 4 °C. The antibodies’ information was listed as follows: γH2AX (rabbit, 1:200, Cell Signaling Technology, Boston, MA, USA, 9718S), H2AX (rabbit, 1:250, Cell Signaling Technology, Boston, USA, 7631T), ATM (rabbit, 1:100, Proteintech, Chicago, IL, USA, 27156-1-AP), TUJ1 (mouse, 1:500, Sigma, Darmstadt, Germany, A5441), GS (mouse, 1:1000, Millipore, Chicago, IL, USA, MAB302), IBA-1 (goat, 1:3000, Abcam, Cambridge, UK, ab5076), FABP7 (guinea pig, 1:500, Asisbiofarm, Hangzhou, China, OB-PGP011-01), NF200 (mouse, 1:1000, Millipore, Chicago, IL, USA, MAB377), CGRP (goat, 1:500, Bio-Red, 1720-9007), IB4 conjugate (1:70, Sigma, Darmstadt, Germany, L2140), IB4-FITC (1:70, Sigma, Darmstadt, Germany, S3762), NeuN (mouse, 1:1000, Millipore, Chicago, IL, USA, MAB377), and GFAP (mouse, 1:500, Millipore, Chicago, IL, USA, MAB360). The slices were then incubated at room temperature with different fluorescein-labeled FITC and Cy3 secondary antibodies (1:1000, Jackson, West Grove, PA, USA) for 2 h. For IB4 immunohistochemistry, matching IB4-FITC was diluted with 5% skim milk to conduct double staining. 

For the ND7/23 cells’ immunocytochemistry, after being washed with PBS, the cells were fixed with 4% paraformaldehyde for 10 min. After being washed with PBS again, the cells were sealed with 1% BSA containing 0.1% Triton × 100 for 1 h and then incubated with anti-γH2AX primary antibody and secondary antibodies, in the same process as that conducted for the tissue sections’ immunofluorescence. Finally, 4′,6-diamidino-2-phenylindole (DAPI, 0.1 μg/mL, Sigma, Darmstadt, Germany, 28718-90-3) was added for 5 min at room temperature to stain the nucleus. 

A Nikon Ni-E microscope (Nikon, Tokyo, Japan) and a Leica SP8 confocal microscope (Leica, Wetzlar, Germany) were used to examine the stained sections and capture images. The images were analyzed using ImageJ 2.1.0/1.53c software.

### 4.6. Western Blot

The L3–L5 DRG tissue was separated and homogenized in RIPA cleavage buffer (P0013, Beyotime, Shanghai, China) containing protease and phosphatase inhibitors (4693132001, Roche, Germany). The protein concentration was determined using a BCA analysis kit (23225, Thermo Fisher Scientific, Waltham, MA, USA). The 30μg protein samples were loaded into the SDS–PAGE gel sample pore and then transferred to the 0.22 μm PVDF membrane. Then, the membrane was blocked with 5% no-fat milk and incubated overnight at 4 °C with antibodies against γH2AX (1:500), H2AX (1:500), H3 (rabbit, 1:200, Cell Signaling Technology, Boston, MA, USA, 7631), ERK (rabbit, 1:1000, Cell Signaling Technology, Boston, MA, USA, 9102S), pERK (rabbit, 1:1000, Cell Signaling Technology, Boston, MA, USA, 9101S), AKT (rabbit, 1:1000, Cell Signaling Technology, Boston, MA, USA, 4691), and pAKT (rabbit, 1:1000, Cell Signaling Technology, Boston, MA, USA, 4060). The images were captured using an Odyssey Imaging System (LI-COR Bioscience, Lincoln, Northeast, Dearborn, MI, USA). ImageJ software was used to analyze the intensity of the selected strip.

### 4.7. Real-Time Quantitative PCR

The total RNA of the DRG was extracted using Trizol reagent (Invitrogen in Carlsbad, CA, USA), as previously mentioned in [44]. The quantitative polymerase chain reaction (qPCR) was analyzed using an AceQ qPCR SYBR^®^ Green Master Mix (Vazyme, Nanjing, China) and a real-time detection system (StepOnePlus^TM^, Thermo Fisher Scientific, Waltham, MA, USA). The primer sequences used in this study are listed in Table 1. The PCR amplifications were performed at 95 °C for 3 min, followed by 40 cycles of thermal cycling at 95 °C for 10 s and 60 °C for 30 s. The relative expression level for each target gene was normalized using the method of 2^−ΔΔCt^.

### 4.8. Preparation of DRG Neurons and Electrophysiological Recording

The DRG neurons were isolated and cultured as described previously in [43]. In brief, 4–6-weeks-old C57BL/6 mice were deeply anesthetized using isoflurane. After decapitation, the spine was removed and cut open along the midline of the spine. The lumbar DRG (L3–L5) was carefully digested for 30 min in digestive juice containing collagenase and disperse enzyme. To discard the supernatant, 1000 rpm centrifugation for 3 min at room temperature was used. The cells were resuscitated in DRG complete medium (87% Neurobasic medium, 10% Australian fetal bovine serum, 2% Bray 27 plus, 1% PS and 0.25% Glutmax) and cultured overnight at 37 °C in the 5% carbon dioxide cell culture incubator.

The patch pipettes with suitable resistance (4–8 MΩ) were drawn using a P-97 Flaming micropipette puller (Sutter Instruments, Novato, CA, USA) and injected into the electrode solution (121 mM of potassium gluconate, 20 mM of KCl, 0.2 mM of EGTA, 4 mM of Na2ATP, 0.4 mM of GTP-Tris, 2 mM of MgCl2, and 10 mM of HEPES). Neurons with a uniform cytoplasm were selected using an infrared-differential interference contrast microscope (BX51WI, Olympus, Tokyo, Japan), and whole-cell patch clamp recording was performed using a dual-channel patch clamp amplifier (EPC10 USB). The action potential (AP) was induced by a series of ramp currents (current intensity: 100 pA~300 pA; duration: 1 s). PatchMaster software v2×80 was used for the data acquisition (HEKA), and Clamp fit (version 8.0, Origin Lab, Northampton, MA, USA) was used for the data analysis. 

### 4.9. Quantification and Statistics

Mean ± SEM was used to analyze all of the statistical data. GraphPad Prism 8 was used to conduct all of the statistical analyses. The molecular, biochemical, and morphological results were analyzed using a two-tailed Student’s *t*-test and a one-way ANOVA followed by a Bonferroni test. The behavioral data were analyzed using a two-way ANOVA followed by a Bonferroni test. *p* < 0.05 was considered to be statistically significant. 

## 5. Conclusions

This study demonstrates that γH2AX in DRG is down-regulated after SNI, and that this is caused by the down-regulation of H2AX and ATM. The inhibition of ATM promotes neuropathic pain, and the knockdown of Pp2a alleviates SNI-induced neuropathic pain. The results are somewhat preliminary but nevertheless promising, and more research on this matter is needed, which could perhaps shed new light on neuropathic pain research, diagnosis, and treatment.

## Figures and Tables

**Figure 1 ijms-24-10148-f001:**
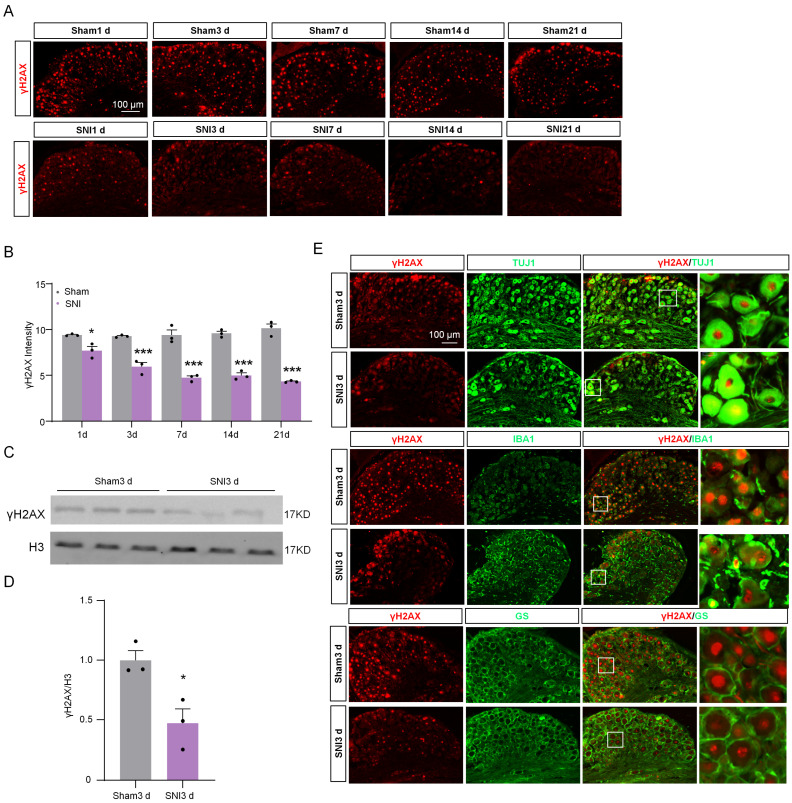
Immunofluorescence was used to detect the distribution and expression of γH2AX in DRG after SNI or sham operation. (**A**) The representative immunofluorescence images of γH2AX expression in the DRG from sham- and SNI-operated mice. Scale bar = 100 μm. (**B**) The statistics show the γH2AX intensity in the ipsilateral DRG after SNI or sham surgery. n = 3 sections per mouse from 3 mice per group. *, *p* < 0.05, ***, *p* < 0.001 versus the corresponding sham group. Two-tailed unpaired Student’s *t*-test was used. (**C**,**D**) Expression of γH2AX protein in the ipsilateral DRG after SNI or sham surgery. n = 3 mice per group. *, *p* < 0.05 versus the sham group by two-tailed unpaired Student’s *t*-test. (**E**) Representative immunofluorescence images for γH2AX and the macrophage marker IBA1, the neuronal marker TUJ1, and the satellite marker GS in DRG from sham and SNI-operated mice. Scale bar = 100 μm. The box area of the merged image is enlarged on the right.

**Figure 2 ijms-24-10148-f002:**
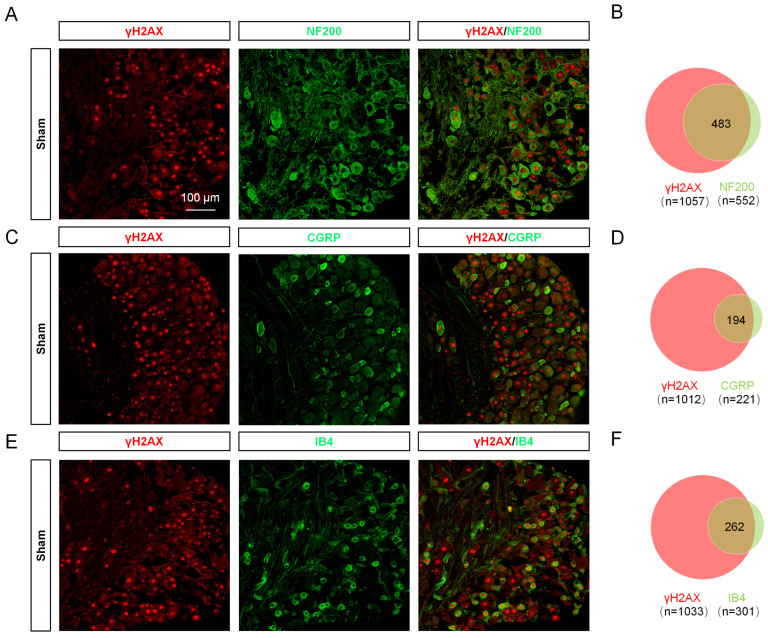
γH2AX was distributed in large peptidergic, small peptidergic, and nonpeptidergic neurons in different proportions. (**A**,**C**,**E**) Immunofluorescent images show the colocalization of γH2AX with NF200 (large myelinated neuron marker), CGRP (small peptidergic neuron marker), and IB4 (small nonpeptidergic neuron marker) in the DRG of sham-operated mice. Scale bar = 75 μm. (**B**,**D**,**F**) Venn diagrams showing the double staining of γH2AX with IB4, CGRP, and NF200.

**Figure 3 ijms-24-10148-f003:**
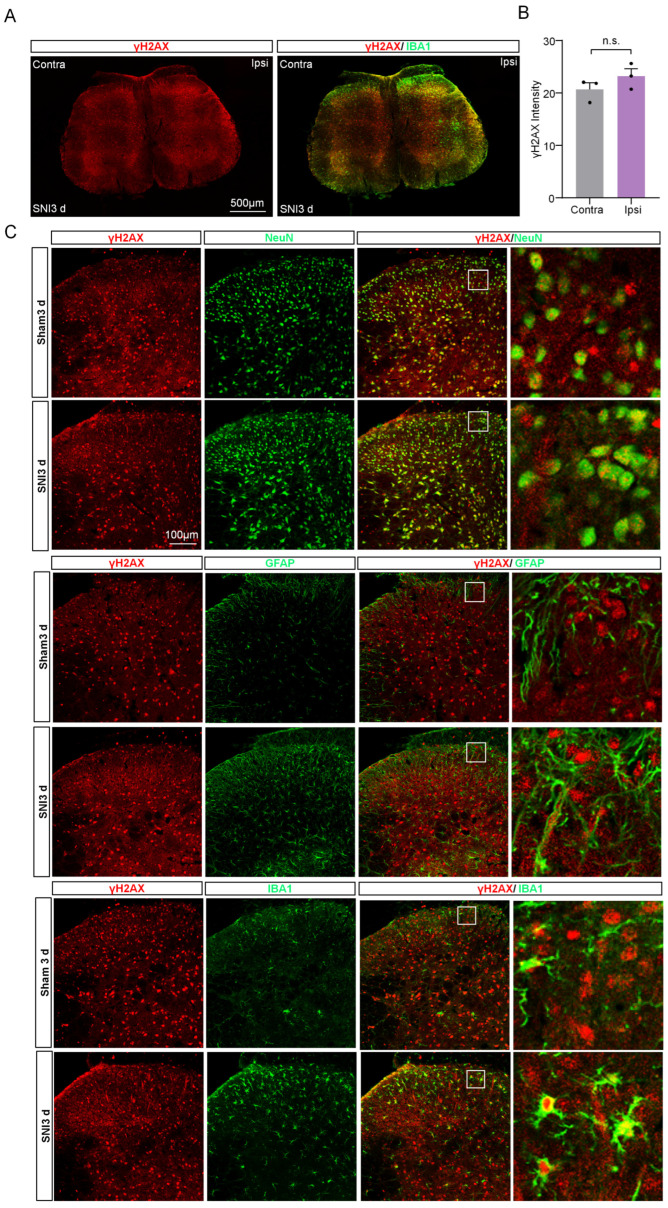
The expression and distribution of γH2AX in the spinal cord after SNI. (**A**) The γH2AX with IBA1 immunofluorescence staining in the spinal cord 3 days after SNI. Scale bar = 100 μm. (**B**) Statistical analysis shows the γH2AX intensity in the ipsilateral spinal dorsal horn and the contralateral side at three days after SNI. Contra, contralateral. Ipsi, ipsilateral. n = 3 sections per mouse from 3 mice per group. Two-tailed Student’s *t*-test was used. *p* = 0.2559. n.s., not significant. (**C**) Representative photomicrographs show double fluorescence labeling directed to γH2AX/NeuN, γH2AX/GFAP, and γH2AX/IBA-1 in the SC. Scale bar = 100 μm. The boxed region of the merged images was enlarged on the right.

**Figure 4 ijms-24-10148-f004:**
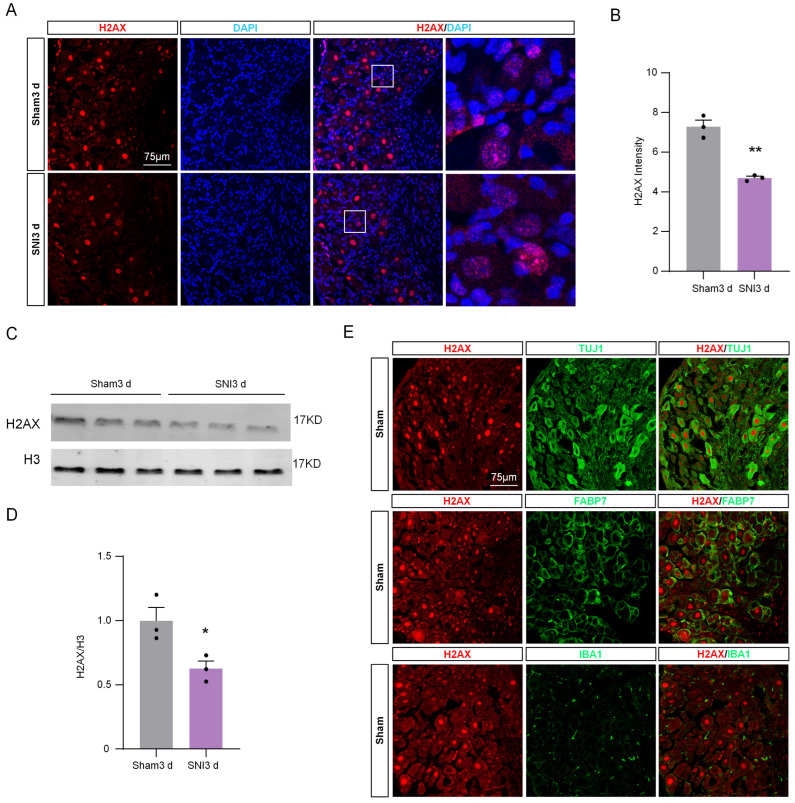
The expression of H2AX in DRG was down-regulated after SNI. (**A**) Representative immunofluorescence images of H2AX (red) and DAPI (blue) in DRG after sham or SNI surgery. Scale bar = 75 μm. The boxed region of the merged images was enlarged on the right. (**B**) Quantification of the fluorescence intensity of H2AX in DRG after SNI or sham surgery. n = 3 sections per mouse from 3 mice per group. **, *p* < 0.01 versus the sham group using two-tailed unpaired Student’s *t*-test. (**C**) Expression of H2AX protein in the DRG of sham and SNI mice. (**D**) Quantification of H2AX levels in (**C**). n = 3. *, *p* < 0.05 versus the sham group using two-tailed unpaired Student’s *t*-test. (**E**) Representative immunofluorescence images for H2AX and the neuronal marker TUJ1, the macrophage marker IBA1, and the satellite marker FABP7 in DRG from sham and SNI operated mice. Scale bar = 75 μm.

**Figure 5 ijms-24-10148-f005:**
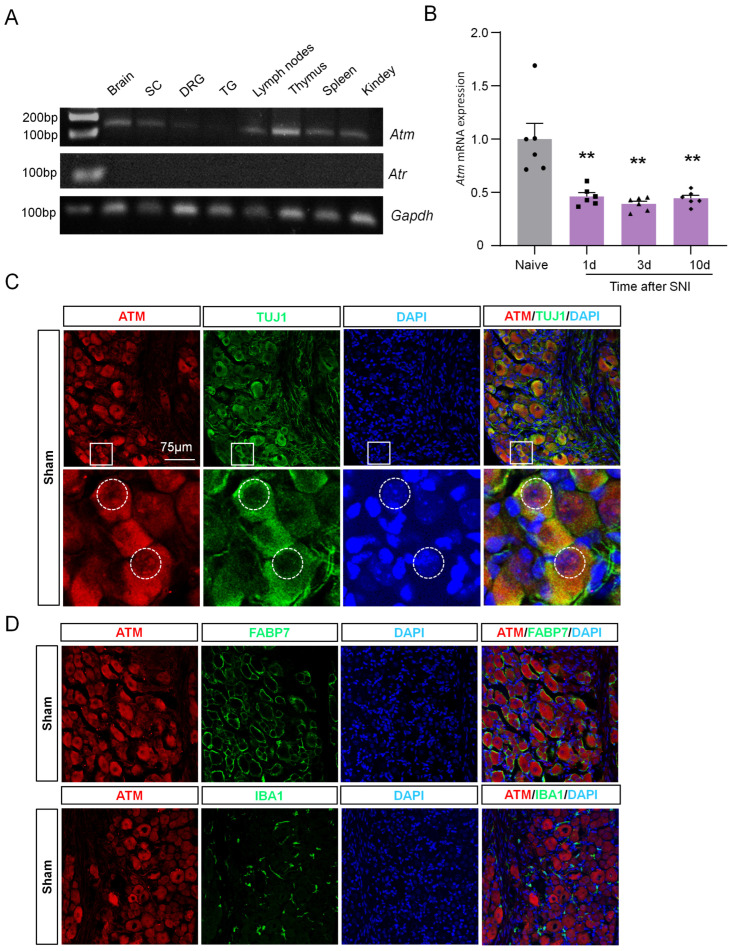
ATM expression was down-regulated in DRG neurons after SNI. (**A**) Gene expression profile of *Gapdh*, *Atm*, and *Atr* in brain, SC, DRG, trigeminal ganglion (TG), lymph nodes, thymus, spleen, and kidney. After 32 cycles of qPCR, the products of *Gapdh*, *Atm,* and *Atr* were detected using agarose electrophoresis. (**B**) Time-course of *Atm* mRNA expression in the DRG from naïve and SNI-operated mice. n = 6 mice per group. **, *p* < 0.01 versus the naive group. One-way ANOVA was followed by Bonferroni’s test. (**C**) Representative immunofluorescence images of ATM and TUJ1 with DAPI after sham surgery. The boxed region of the merged images was enlarged below. The dotted box selects the edge of the DAPI. Scale bar = 75 µm. (**D**) Representative immunofluorescence images of ATM and FABP7, and IBA1 with DAPI, after sham surgery. Scale bar = 75 µm.

**Figure 6 ijms-24-10148-f006:**
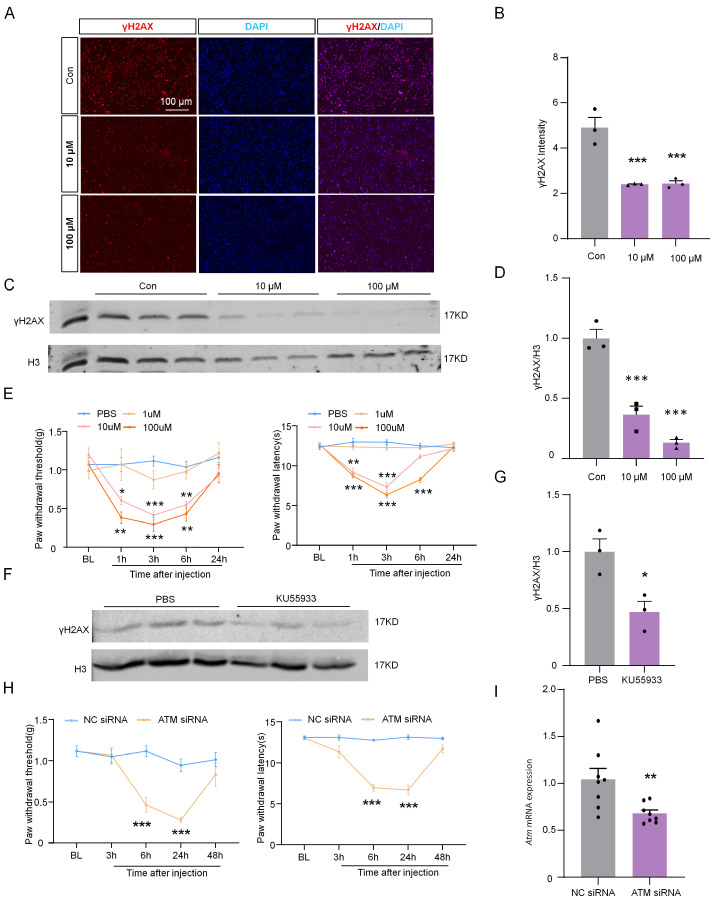
ATM inhibitor KU55933 deregulated γH2AX and induced neuropathic pain. (**A**) Representative images of γH2AX (red) and DAPI (blue) in ND7/23 after 10 and 100 μM of KU55933 were incubated for 6 h. Scale bar = 100 µm. (**B**) The intensity of γH2AX in the ND7/23 after KU55933 was incubated for 6 h. n = 3. ***, *p* < 0.001 versus the control group. One-way ANOVA followed by Bonferroni’s test. (**C**) The expression of γH2AX protein in ND7/23 was cultured with 10 and 100 μM of KU55933 for 6 h. (**D**) Quantification of γH2AX levels in (**C**) n = 3. ***, *p* < 0.001 versus the control group. One-way ANOVA was followed by the Bonferroni’s test. (**E**) Effect of intrathecal injection with PBS or KU55933 (1, 10, and 100 μM) on paw withdrawal responses to mechanical and heat stimuli. n = 8. *, *p* < 0.05. **, *p* < 0.01. ***, *p* < 0.001 versus the PBS group. Two-way ANOVA was followed by Bonferroni’s test. (**F**) The expression of γH2AX protein in DRG after intrathecal injection with PBS or KU55933 (100 μM) for 6 h. (**G**) Quantification of γH2AX levels in (**F**). n = 3. *, *p* < 0.05 versus the PBS group. Two-tailed unpaired Student’s *t*-test was used. (**H**) Effect of naive ICR mice intrathecal injection with PP2A siRNA (2 µg) or NC siRNA (2 µg) on paw withdrawal responses to mechanical and heat stimuli. n = 8 mice per group. NC, negative control siRNA. ***, *p* < 0.001 compared with the NC siRNA group. Two-way ANOVA was followed by post hoc Bonferroni’s test. (**I**) The mRNA expression of *Atm* after intrathecal injection with ATM siRNA or NC siRNA for 24 h. **, *p* < 0.01 compared with the NC siRNA group, n = 8. Two-tailed unpaired Student’s *t*-test was used.

**Figure 7 ijms-24-10148-f007:**
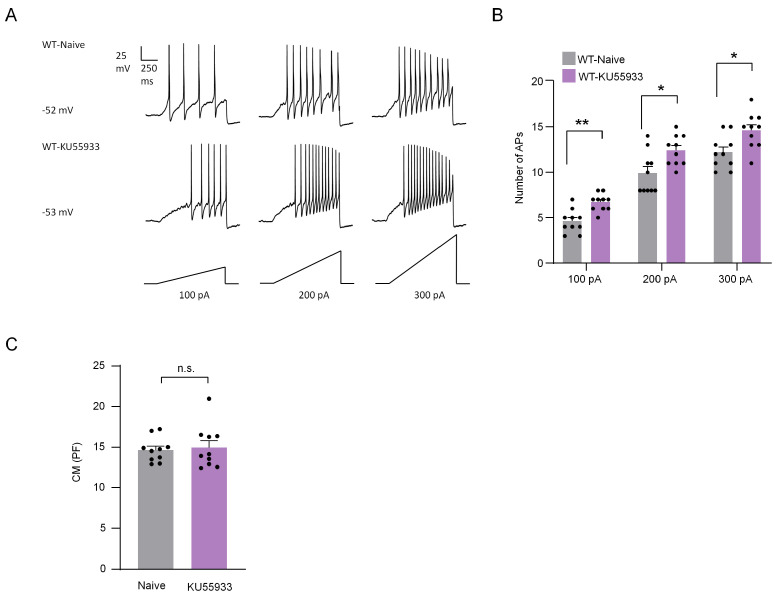
ATM inhibitor KU55933 increased the action potentials in DRG neurons. (**A**) Examples of membrane potential responses evoked by 1000 ms current injection of 100 pA, 200 pA, and 300 pA in naive and KU55933 treated neurons from WT mice. (**B**) Histogram showing a significant increase in the numbers of APs in DRG neurons pretreated with KU55933 compared to naive group. n = 10 neurons/group. *, *p* < 0.05. **, *p* < 0.01 compared with the naive group. Two-way ANOVA was followed by the Bonferroni test. (**C**) There was no difference between the two groups in CM. Two-tailed unpaired Student’s *t*-test was used. n.s., not significant.

**Figure 8 ijms-24-10148-f008:**
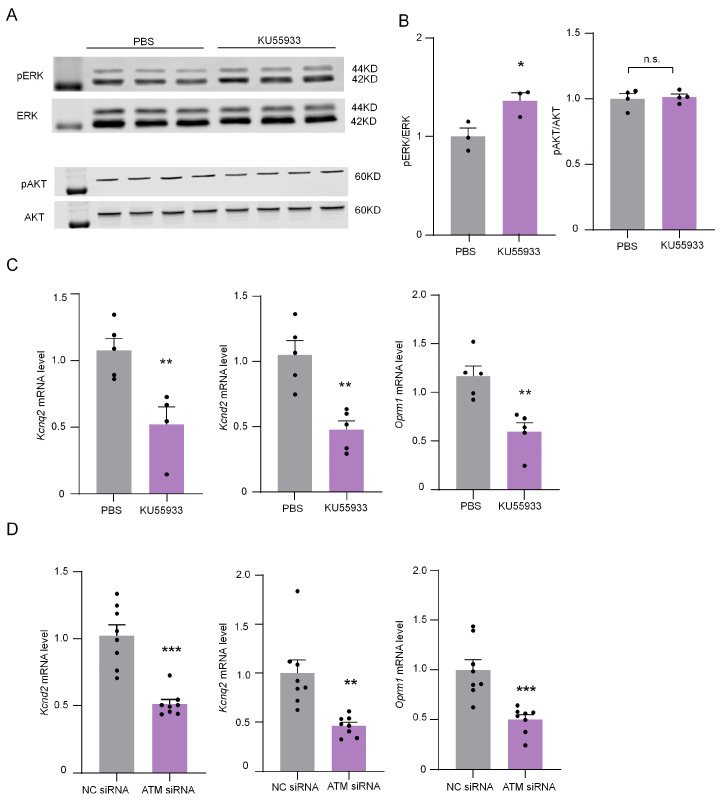
ATM inhibitor KU55933 influenced the ERK and AKT phosphorylation levels and down-regulated the mRNA expression of *Kcnd2*, *Kcnq2*, and *Oprm1*. (**A**) The expression of pERK, ERK, pAKT, and AKT in DRG after intrathecal injection of PBS or KU55933 (100 μM) for 6 h. (**B**) Quantification of pERK and pAKT levels in (**A**). n = 3. *, *p* < 0.05 compared with the PBS group. Two-tailed unpaired Student’s *t*-test was used. n.s., not significant. (**C**) The mRNA expression of *Kcnq2*, *Kcnd2,* and *Oprm1* after applying PBS or KU55933 (100 μM) for 6 h. n = 5. **, *p* < 0.01 compared with the PBS group. Two-tailed unpaired Student’s *t*-test was used. (**D**) The mRNA expression of *Kcnq2*, *Kcnd2*, and *Oprm1* after applying ATM siRNA or NC siRNA for 24 h. n = 8. **, *p* < 0.01, ***, *p* < 0.001 compared with the PBS group. Two-tailed unpaired Student’s *t*-test was used.

**Figure 9 ijms-24-10148-f009:**
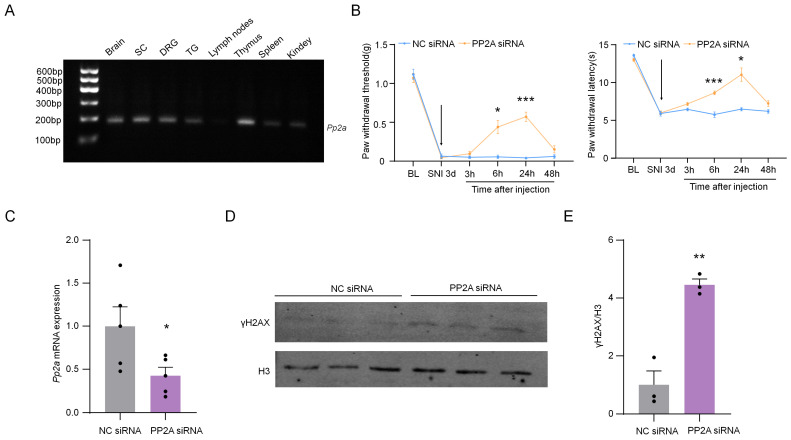
Intrathecal injection of PP2A siRNA can effectively relieve pain induced by SNI. (**A**) Gene expression profile of *Pp2a* in the brain, SC, DRG, TG, lymph nodes, thymus, spleen, and kidney. After 32 cycles of qPCR, the products of *Pp2a* were detected using agarose electrophoresis. (**B**) Effect of intrathecal injection with PP2A siRNA (2 µg) or negative control (NC) siRNA (2 µg) on paw withdrawal responses to mechanical and heat stimuli. n = 7. *, *p* < 0.05, ***, *p* < 0.001 compared with the NC siRNA group. Two-way ANOVA was followed by Bonferroni’s test. The arrow means the time of siRNA injection. (**C**) The mRNA expression of *Pp2a* in DRG from SNI-operated mice after intrathecal injection with PP2A siRNA or NC siRNA for 24 h. *, *p* < 0.05 compared with the NC siRNA group, n = 5. Two-tailed unpaired Student’s *t*-test was used. (**D**) The expression of PP2A protein in DRG from SNI-operated mice after intrathecal injection with PP2A siRNA or NC siRNA for 24 h. (**E**) Quantification of PP2A levels in (**D**). n = 3. **, *p* < 0.01 compared with the NC siRNA group. Two-tailed unpaired Student’s *t*-test was used.

**Table 1 ijms-24-10148-t001:** qPCR primer sequences.

Primer	Sequence (5′ to 3′)	Amplicon Size
*Kcnd2* forward	CGT GCC TGT GAT CGT GTC	161 bp
*Kcnd2* reverse	TGC TCA GTA GCC CAT TCC	
*Kcnq2* forward	CGT GAC TAT CGT GGT ATT CGG	150 bp
*Kcnq2* reverse	GCA ATG GAG GCA ATC AGC	
*Oprm1* forward	GCC TTA GCC ACT AGC ACG	191 bp
*Oprm1* reverse	AAA TCC AGG GCC TTG ACC	
*Atm* forward	GAG CGT CTC AAG ATA ACC C	139 bp
*Atm* reverse	CCT ATT TCT CCC AAA CAC C	
*Atr* forward	CCC AAA TAG CAA GGA ATA A	100 bp
*Atr* reverse	AAG AGT GCG AAA GGT ATC A	
*Gapdh* forward	GTA AGA AAC CCT GGA CCA CCC	
*Gapdh* reverse	AGG GAG ATG CTC AGT GTT GG	89 bp

## Data Availability

The data involved in this study can be obtained by contacting the corresponding authors.

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
