# Peer review of "Nerve Injury-Induced γH2AX Reduction in Primary Sensory Neurons Is Involved in Neuropathic Pain Processing"

_ijms, 2023, doi:10.3390/ijms241210148_

Round 1

Reviewer 1 Report

ijms-2393930

The authors submitted a manuscript entitled „ Nerve injury-induced γH2AX reduction in primary sensory 2 neurons involved in neuropathic pain processing".

Major concern:

This is very surprising and not supported by current literature or by the presented data, because it is well established that injured neurons display fragmented DNA as reflected by increased DNA protein markers such as γH2AX. Indeed, various types of nerve injury such as ischemia/reperfusion (Zhou et al., 2023; Apoptosis 2023; 28:447–457), ischemia-induced retinal neurocytes damage (Yang et al, Cell Death Dis 2016;7:e2473), radiation induced sensory ganglion neurons damage ( Casafont et al., Mol Neurobiol 2016:6799–6808; Acta Neuropathologica  122, 481–493; 2011) and chronic constriction injury-induce neuropathic pain (Chen et al., Int. J. Mol. Sci. 2022, 23, 11974) showed upregulation of mRNA, proteins and immunorectivity of DNA damage marker γH2AX.

Moreover, the scientific validity of data is questionable when in almost every experiments the number of data is n=3, apparently without any power analysis?

Throughout the manuscript the authors suddenly refer to results from ND7/23 cultured cells, however, it is not mentioned anywhere in the manuscript which experiments were performed?

Introduction

In many cases, the authors did not support their claims with enough references.

Material and Methods

The Method section is lacking many details.

Results

Beta tubulin III, also known as Tuj-1 is exclusively in neurons and used as general neuronal marker including sensory neurons, therefore, using the term sensory neuron not correct.

The results are poorly presented and lacking explanation of figures. More details are required in the figure legends.

The figure 1, 2, and 3 have bad quality and are not convincing because all presented DRG pictures are not sharp and double immunofluorescence of γH2AX with IB4-, CGRP or RT200 does not add any information to the manuscript because of bad quality pictures with the specific regard to CGRP staining; in addition, no quantification of ratio of γH2AX with neuronal subtypes was performed.

In Fig. 4 the picture shows bad quality and is not convincing to be used for semi-quantification.

Fig. 5B very bad quality because ATM staining is not specific and I can see only background without any specific immunoreactivity. In addition, western blot in Fig.5D is not convincing.

In Fig. 9D western blot is not convincing.

Discussion

The discussion is poorly written and needs considerable improvement. The authors did not critically discuss and interpret their results in light of previously published studies.

Reviewer 2 Report

This manuscript deals with epigenetic regulation in chronic pain. To that end, they investigated γH2AX as a DNA damage marker that regulates DNA double-strand breaks. They found the expression of γH2AX decreased in mouse DRG neurons after nerve injury and unchanged in the spinal cord. ATM, that promotes H2AX phosphorylation was also down-regulated in DRG. Intrathecal injection of ATM inhibitor KU55933 downregulated γH2AX expression and significantly induced mechanical allodynia and thermal hyperalgesia in a dose-dependent manner. Further, the inhibition of dephosphorylation of γH2AX partially suppressed this down-regulation and relieved pain. Their results indficate that down-regulation of γH2AX may contribute to neuropathic pain by downregulating inhibitory genes.

The study was well designed and manuscript to the most part clearly written and nicely illustrated. Before it gets published, there are several remarks from my side:

1. The title "Nerve injury-induced γH2AX reduction in primary sensory neurons involved in neuropathic pain processing" seems inprecise, also gramatically incorrect (should be "is involved in..."). However, I find "involved in" a vague qualification, as the results clearly suggest that γH2AX reduction increases neuropathic pain, so it could be stated in the title.

2. Iba-1 is marker for both macrophages and microglia in the CNS (spinal cord), so it should be clarified in the 2.2. subchapter of results.

3. Although largely celarly written, there are examples of gramatically incorrect sentences throughout the manuscript, especially in the Methods section, e.g. "Use the 38 Nikon Ni-E microscope (Nikon, Tokyo, Japan) to examine stained sections and capture images." (pg.13, ln 38-39). The use of plural "tissues" means "tissue wipes", DRG tissue in histological sense is not a plural, etc, etc... I would suggest a careful proofreading of the text for such errors.

Although largely celarly written, there are examples of gramatically incorrect sentences throughout the manuscript, especially in the Methods section, e.g. "Use the 38 Nikon Ni-E microscope (Nikon, Tokyo, Japan) to examine stained sections and capture images." (pg.13, ln 38-39). The use of plural "tissues" means "tissue wipes", DRG tissue in histological sense is not a plural, etc, etc... I would suggest a careful proofreading of the text for such errors.

Round 2

Reviewer 1 Report

the revised manuscript has been extensively improved